# Uncertainty-aware Fine-tuning for Time Series Anomaly Detection

## Abstract

Time-series anomaly detection is a crucial task in various real-world domains, geared towards identifying data observations that significantly deviate from the norm. Although time-series foundation models have shown promising results across multiple tasks, their effectiveness in anomaly detection is often inferior. This is due to their unsupervised learning paradigm being compromised by anomaly contamination in the training data. In addition, the existing approaches lack the capability to capture boundries of multiple types of normal and abnormal patterns. To overcome these challenges, we propose *ULoRA-MoE*, a general uncertainty-aware fine-tuning approach using resource-efficient Mixture-of-Expert (MoE) module based on LoRA. This proposed approach can enhance the fine-tuning performance across a broad spectrum of time series foundation models for anomaly detection. Each expert module of MoE can help learn different types of anomalies. Furthermore, we design the uncertainty-aware router of MoE using Gumbel-Softmax distribution for categorical sampling to capture the epistemic uncertainty. Given the estimated uncertainty, we propose a calibrated anomaly score function to mitigate the detrimental effects of anomaly contamination. We conducted extensive experiments on two general types of time series foundation models. The results demonstrate that our approach significantly improves the model performance compared to existing fine-tuning approaches. Furthermore, *ULoRA-MoE* shows competitive performance compared to a comprehensive set of non-learning, classical learning, and deep learning (DL) based time-series anomaly detection baselines across 8 real-world benchmarks.

## 1 Introduction

Over recent decades, the exponential growth in informatization has generated vast amounts of time series data. These data streams, sourced from systems such as large-scale data centers, cloud servers, and spacecraft, are invaluable for monitoring and detecting potential faults, threats, and risks by identifying unusual states (i.e., anomalies) (Cook et al., 2019; Anandakrishnan et al., 2018; Kieu et al., 2022a). Anomaly detection, a key field in data mining and analytics, focuses on identifying exceptional data observations that deviate significantly from the norm (Pang et al., 2021). This capability is crucial for ensuring the reliability and safety of various target systems. Given the high cost and difficulty of labeling in real-world applications, time series anomaly detection is typically formulated as an unsupervised task with unlabeled training data.

Foundation models, trained on extensive and diverse datasets, have revolutionized several research areas by supporting a wide range of tasks with minimal additional training (Bommasani et al., 2021). They have significantly impacted fields such as language modeling (Touvron et al., 2023; Achiam et al., 2023; Brown et al., 2020) and computer vision (Liu et al., 2024; Kirillov et al., 2023). Recent advancements in time series analysis have aimed to develop models with similar capabilities, creating novel architectures capable of capturing diverse time series signals. Notable examples include UniTS (Gao et al., 2024b) and the Moment model (Goswami et al., 2024), which are built on the transformer backbone. Another popular trend is to reprogram the large language models (LLMs) for time series tasks, as demonstrated by models such as GPT4TS (Zhou et al., 2023) and Time-LLM (Jin et al., 2023). These models, either pretrained on diverse time series datasets or utilizing pretrained language models, are versatile and can be applied to multiple time series tasks, including anomaly detection.

However, the fine-tuning schemes for time series foundation models in anomaly detection tasks remain underexplored. Traditional fine-tuning approaches often assume that the entire training set consists of normal samples, which is not always the case. Training datasets may be contaminated by anomalies, leading to significant disruptions in the learning process and severe overfitting issues (Xu et al., 2024). Moreover, without knowledge of anomalies, models trained solely on normal data may have difficulty accurately exploring the boundaries between normal and various types of anomalous data. To address these challenges, we introduce an uncertainty-aware fine-tuning framework *ULoRA-MoE* for time series foundation models to improve the anomaly detection performance. *ULoRA-MoE* utilizes resource-efficient Mixture-of-Expert (MoE) module based on LoRA to learn different types of anomalies and accurately define the boundary between each type of the normal and anomalous data. Furthermore, we use Gumbel-Softmax distribution for categorical sampling on the router of MoE to estimate the epistemic uncertainty of the foundation model. Given the estimated uncertainty, we design an uncertainty-based anomaly score function to calibrate the foundation model with respect to the contaminated training data, thereby helping to mitigate the detrimental effects of anomaly contamination. Our contributions are summarized as follows:

- *ULoRA-MoE*: We introduce an uncertainty-aware Mixture-of-Expert fine-tuning approach based on LoRA, tailored for two general types of time series foundation models in anomaly detection tasks.

- Probabilistic reconstruction for uncertainty quantification: We design the Gumbel-Softmax sampling approach on the MoE router to estimate the epistemic uncertainty of the time series foundation model,

- Calibrated anomaly score function: We propose a calibrated anomaly score function to mitigate the detrimental effects of anomaly contamination.

- Empirical Validation: We demonstrate that *ULoRA-MoE* achieves state-of-the-art performance across 8 real-world time series anomaly detection benchmarks.

## 2 PRELIMINARIES

### 2.1 PROBLEM FORMULATION

We introduce notations to formally define the unsupervised time-series anomaly detection task. The training data consist of a multivariate time series $\mathcal{X} = [\mathbf{x}_1, \ldots, \mathbf{x}_T] \in \mathbb{R}^{T \times F}$, where $T$ is the number of timestamps and $F$ is the feature dimension. The test set is denoted as $\hat{\mathcal{X}} = [\hat{\mathbf{x}}_1, \ldots, \hat{\mathbf{x}}_T] \in \mathbb{R}^{\hat{T} \times F}$ with labels $\hat{\mathbf{y}} = [\hat{y}_1, \ldots, \hat{y}_T] \in \{0, 1\}^{\hat{T}}$, where $\hat{y}_t = 0$ indicates a normal timestamp and $\hat{y}_t = 1$ indicates an anomaly. The goal is to learn a score function $f_\theta : \mathcal{X} \to \mathbb{R}$ such that $f_\theta(\mathbf{x}_t) = \tilde{y}_t$ estimates the anomaly value $\hat{y}_t$. The parameters $\theta$ are estimated using the training data $\mathcal{X}$.

### 2.2 LORA

The Low-Rank Adaptation (LoRA) technique Hu et al. (2021) modifies a pre-trained model by freezing its original weights, denoted as $W_0$, and introducing updates through a low-rank decomposition. This can be expressed mathematically as:

$$W = W_0 + \Delta W = W_0 + B \cdot A \tag{1}$$

where $\{W, W_0, \Delta W\} \in \mathbb{R}^{k \times d}$. $B \in \mathbb{R}^{k \times r}$ and $A \in \mathbb{R}^{r \times d}$ represent two trainable low-rank matrices. The rank $r$ is significantly smaller than both $d$ and $k$. This constraint ensures that the updates $\Delta W$, comprised of the product $B \cdot A$, remain low-rank. The matrices $W$ and $\Delta W$ are then multiplied with the same input $\mathbf{x}$, resulting in a modified forward pass expressed as:

$$\mathbf{h} = W_0 \cdot \mathbf{x} + \Delta W \cdot \mathbf{x} = W_0 \cdot \mathbf{x} + B \cdot A \cdot \mathbf{x} \tag{2}$$

where $\mathbf{h}$ is the hidden representation. This equation highlights how both the original and the updated components contribute to the model's output, facilitating significant model adaptation with minimal changes to the parameter space.

## 2.3 LoRA Mixture of Experts (MoE)

LoRA Mixture-of-Experts (MOE) (Li et al., 2024; Wu et al., 2024b) has been designed for multi-task learning with LLMs. It is an efficient approach to scale the number of parameters while maintaining the same computational bounds. It incorporate LoRA into MoE involves applying low-rank updates specifically to the expert networks within the MoE architecture. The LoRA expert network is explicitly designed for each domain and the weights are updated using the following formula:

$$W = W_0 + \textbf{router}_{W^r}(W_1, ..., W_K) = W_0 + \textbf{router}_{W^r}(B_1 A_1, B_2 A_2, ..., B_K A_K) \tag{3}$$

where $W^r \in \mathbb{R}^{hidden\_dim \times K}$, $W \in \mathbb{R}^{k \times d}$, $B \in \mathbb{R}^{k \times r}$ and $A \in \mathbb{R}^{r \times d}$. $B_k A_k$ defines the LoRA module $\mathbf{E}_k$, which is repeated multiple times within each transformer layer to reduce trainable parameters. The goal is to adapt each of the LoRA module $\mathbf{E}_k$ to different domain tasks. $W^r$ denotes the learnable parameter of the router. Given the hidden representation $h \in \mathbb{R}^{hidden\_dim}$ and $W_k^r$, the gate probability for routing the hidden representation $h$ to LoRA module $\mathbf{E}_k$ is denoted as:

$$\alpha(\mathcal{M}_k) = \frac{\exp(\mathbf{h} \cdot W_k^r)}{\sum_{j=0}^{K} \exp(\mathbf{h} \cdot W_j^r)}. \tag{4}$$

The routers from previous methods are usually deterministic. For example, they select and activate $k$ experts using top-$k$ gated values.

## 3 Methodology

We propose a fine-tuning approach for anomaly detection within general time series foundation models using reconstruction-based self-supervised learning. Our goal is to fine-tune the pretrained model $\mathcal{M}$ utilizing unlabeled fine-tuning data $\mathcal{X}$. We aim to develop representations that clearly distinguish between normal patterns and anomalies. We hypothesize that anomalies in time series data typically manifest as rare and inconsistent behaviors, which often result in less confident predictions by the model during the reconstruction phase. By leveraging epistemic uncertainty, we seek to lessen the effects of anomaly contamination, thus enhancing the model calibration against the contaminated fine-tuning data. This approach necessitates answering two key questions: *How can we efficiently quantify epistemic uncertainty across various pretrained models?* and *How can we leverage the identified uncertainty to calibrate the fine-tuning process?* To address the first question, we introduce a sampling-based routing strategy to effectively measure the epistemic uncertainty and present the *ULoRA-MoE* framework in 3.1 and 3.2. For the second question, we develop an uncertainty-aware anomaly scoring method detailed in 3.3.

### 3.1 Sampling-based routing strategy

To efficiently quantify the epistemic uncertainty across a diverse set of pretrained models with transformer architectures, we propose a sampling-based routing strategy. This strategy leverages the LoRA MoE model architecture, allowing us to sample from selecting a few LoRA experts at each layer with LoRA module instead of the entire parameter space of the time series foundation model. This approach significantly reduces memory usage and computation time, making efficient uncertainty quantification possible on large pretrained foundation models. Additionally, this strategy does not necessitate modifications to the existing modules of the pretrained model, enhancing its flexibility for application to any transformer-based time series foundation model.

However, the softmax gates typically used in the original MoE architecture (Shazeer et al., 2017) are not conducive to sampling, as they hinder the calculation of useful gradients for backpropagation. To address this issue, we propose a novel routing mechanism using a Gumbel-Softmax gate (Maddison et al., 2016; Li et al., 2023). Specifically, we define a gate logit $z_k = \mathbf{h} \cdot W_k^r$ for each LoRA expert module $E_k$, where the gate value $g_k$ is sampled from the Gumbel-Softmax distribution during training. The sampling method is defined by the following equation:

$$g_k = \frac{\text{softmax}(z_k + \epsilon)}{\tau} \tag{5}$$

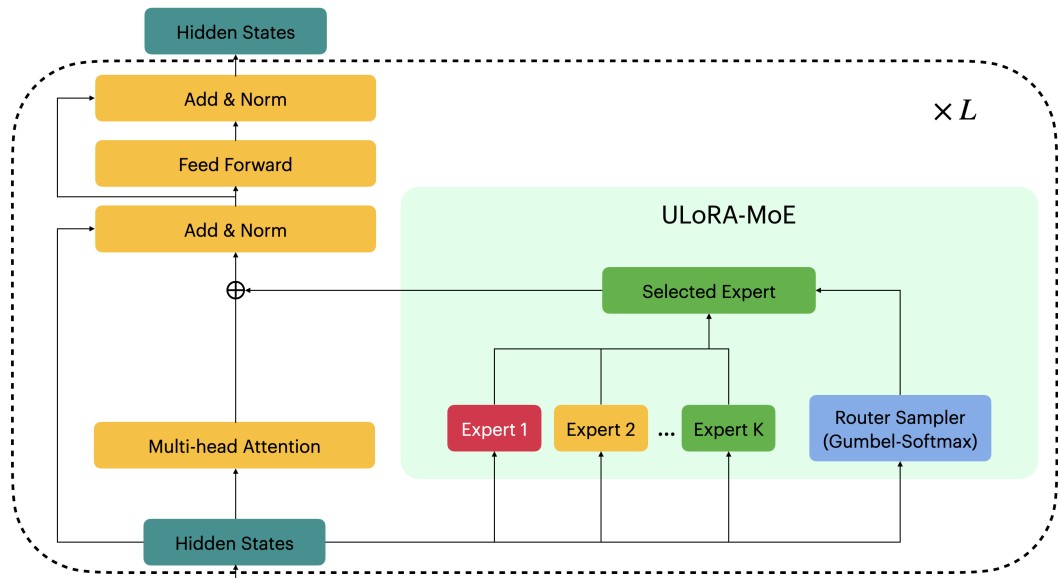

Figure 1: *ULoRA-MoE* model architecture for time-series anomaly detection. The proposed Gumbel-Softmax router sampler is used to select the LoRA expert from $K$ different candidates.

where $\epsilon = -\log(-\log(u))$ represents the Gumbel noise, with $u \sim \text{Uniform}(0, 1)$, and $\tau$ is the temperature parameter that controls the randomness of the distribution. The value of the gate logit $z_k$ can be interpreted as the relative contribution of each expert module in computing the aggregated representation from all experts.

### 3.2  *ULoRA-MoE* FRAMEWORK

In line with the proposed sampling-based routing strategy, we introduce the architecture of *ULoRA-MoE*, depicted in Figure 1. For each layer equipped with a LoRA MoE module, *ULoRA-MoE* initiates with the hidden states as inputs to the current layer, and it subsequently infers $K$ distinct LoRA experts alongside the Gumbel-Softmax router sampler. This sampler plays a crucial role in selecting an expert based on the sampled gate value $g_k$. For the UniTS and GPT4TS pretrained models, we incorporate a PLoRA-MoE module into the multi-head attention layers.

To facilitate the fine-tuning process of *ULoRA-MoE*, we sample once from the Gumbel-Softmax distribution to generate the reconstruction $\hat{\mathbf{x}}$ and employ the mean squared error (MSE) between $\hat{\mathbf{x}}$ and the actual data $\mathbf{x}$ as the reconstruction loss. During inference, we increase the sampling frequency to $T$ times from the Gumbel-Softmax distribution to produce a set of reconstruction samples $\mathbf{x}_{t=1}^{T}$, which are used to form an output distribution for uncertainty quantification. Both the training and inference phases are summarized in Algorithm 1.

### 3.3  UNCERTAINTY-AWARE ANOMALY SCORE FUNCTION

For the second question, we aim to design a score function that balances fidelity of reconstruction and the incorporation of model uncertainty to enhance anomaly detection. This function not only computes the distance between the actual sequence and its reconstruction but also penalizes predictions with high epistemic uncertainty, thereby softly mitigating the influence of anomaly contamination. To achieve this, we use the negative log-likelihood (NLL) as our anomaly score function:

$$\text{NLL} = \frac{1}{2} \log(\sigma^2) + \frac{1}{2\sigma^2}(\mu - \mathbf{x})^2 \tag{6}$$

Here, $\sigma^2$ and $\mu$ are the variance and mean, respectively, derived from the set of $T$ reconstruction samples $\hat{\mathbf{x}}_{t=1}^{T}$. This scoring function ensures that the model not only reconstructs sequences with high

---

**Algorithm 1:** *ULoRA-MoE*

---

**Input:** Fine-tuning dataset $\mathcal{D}_{\text{train}} = \{\mathbf{x}_i\}_{i=1}^M$, test dataset $\mathcal{D}_{\text{test}} = \{\mathbf{x}_i\}_{i=1}^N$, pretrained model
  $p_\theta(x)$, number of experts $K$, number of sampling times $T$.

1 **Phase 1: Fine-tuning**:
2 Fine-tune the pretrained model $p_\theta(x)$ using $\mathcal{D}_{\text{train}}$ with Mean Squared Error as the reconstruction
  loss function.
3 **Phase 2: Inference**:
4 Sample a sequence of experts across all layers equipped with the LoRA MoE module $T$ times.
5 Generate a set of reconstruction samples $\{\hat{\mathbf{x}}_t\}_{t=1}^T$. Fit these reconstruction samples to a Gaussian
  distribution to estimate mean $\mu$ and standard deviation $\sigma$.
6 Compute the Negative Log-Likelihood (NLL) as the anomaly score using Eqn 6.
7 Apply the Streaming Peaks-over-Threshold (SPOT) algorithm (Siffer et al., 2017) to determine
  the anomaly threshold and make anomaly predictions $\{\hat{y}_i\}_{i=1}^N$ on the test dataset.
**Output:** Anomaly predictions $\{\hat{y}_i\}_{i=1}^N$.

---

accuracy but also maintains confidence in its predictions by accounting for both the reconstruction
error and the epistemic uncertainty. Finally, to set a threshold for identifying and labeling anomalies
at each timestamp, we utilize the Streaming Peaks-over-Threshold (SPOT) algorithm (Siffer et al.,
2017), which is based on extreme value theory, to make anomaly predictions on the test dataset.

## 4  RELATED WORK

**Time series anomaly detection.** Time series anomaly detection can be categorized into non-learning,
classical learning, and deep learning methods (Zhao et al., 2022). Non-learning methods encompass
density-based approaches (Breunig et al., 2000; Kriegel et al., 2009; Tang, 2002; Tang et al., 2002)
and similarity-based methods (Yeh et al., 2016). Density-based approaches detect anomalies by
examining the density distribution of data points within clusters, whereas similarity-based methods
identify series that significantly deviate from the majority. Classical learning methods (Liu et al.,
2008) partition time series into windows and use the training set of normal data to classify the
test set based on similarity (Schölkopf et al., 1999). Deep learning methods can be divided into
reconstruction-based and prediction-based approaches. Reconstruction-based methods compress raw
input data and then reconstruct it, where reconstruction errors indicate anomalies (Zhao et al., 2022).
Techniques include autoencoders (AEs) (Krizhevsky et al., 2012; Campos et al., 2021), variational
autoencoders (VAEs) (Park et al., 2018; Li et al., 2021; Su et al., 2019), generative adversarial
networks (GANs) (Zhou et al., 2019; Li et al., 2019; Schlegl et al., 2019), transformers (Chen et al.,
2021; Xu et al., 2021; Yang et al., 2023), and diffusion models (Xiao et al., 2023a; Wang et al.,
2024; Pintilie et al., 2023). Prediction-based methods (Pang et al., 2021) use past observations to
forecast current values and identify anomalies based on prediction errors. Some methods consider the
anomaly contamination problem. Du et al. (2021); Pang et al. (2018; 2020) filter potential anomalies
via self-training. Kieu et al. (2022b) employed an additional autoencoder to clean the dataset before
training. Qiu et al. (2022) jointly infer binary labels while updating model parameters. However,
these filtering processes can misclassify difficult normal samples, which are crucial for training. Xu
et al. (2024) proposes uncertainty modeling-based calibration in a one-class learning objective to
address this, but it relies on specific model architectures and is not applicable for general time series
foundation model fine-tuning.

**Time series foundation models.** Foundation models, trained on extensive and diverse datasets,
support a wide range of tasks with minimal additional training (Bommasani et al., 2021). They have
revolutionized areas such as language modeling (Touvron et al., 2023; Achiam et al., 2023; Brown
et al., 2020) and computer vision (Liu et al., 2024; Kirillov et al., 2023). Recent developments in time
series analysis aim to create models with similar capabilities, capturing diverse time series signals
through novel architectures. For instance, TimesNet (Wu et al., 2022) leverages frequency-based
features derived from Fourier transforms to analyze complex signals. UniTS (Gao et al., 2024b) and
the Moment model (Goswami et al., 2024) utilize transformer architectures, whereas TimeDiT (Cao
et al., 2024) is based on a diffusion transformer structure. These models are pretrained on diverse
time series datasets and can be applied to anomaly detection tasks. Additionally, there is a growing

trend of adapting large language models (LLMs) to serve as foundational models for time series analysis, including LLMTIME (Gruver et al., 2024), LLM4TS (Chang et al., 2023), GPT4TS (Zhou et al., 2023), Tempo (Cao et al., 2023), Time-LLM (Jin et al., 2023) and Lag-Llama (Rasul et al., 2023). Despite the proliferation of these advanced models, the fine-tuning strategies for time series anomaly detection remain underexplored.

**Parameter-efficient Fine-tuning** Instruction fine-tuning enabled large language models (LLMs) to comprehend human intentions and follow instructions, forming the foundation of chat systems (Achiam et al., 2023). However, as the size of these models increases, fine-tuning becomes a time-consuming and memory-intensive process. To address this challenge, various studies have explored different methods: parameter-efficient fine-tuning (PEFT) (Mangrulkar et al., 2022), distillation (Liu et al., 2023b; Xiao et al., 2023b), quantization (Frantar et al., 2022; Xiao et al., 2023c), and pruning (Frantar & Alistarh, 2023; Ma et al., 2023). LoRA (Hu et al., 2021), which employs low-rank matrices to decompose linear layer weights, stands out as one of the most prominent PEFT techniques. It enhances model performance without adding computational overhead during inference and offers a resource-efficient strategy to rapidly adapt LLMs to new tasks with limited training data.

**Mixture-of-Experts** The Mixture of Experts (MoE) is an ensemble approach visualized as a set of sub-modules or 'experts', each tailored to different input data types (Jacobs et al., 1991; Cai et al., 2024). Modern MoE versions enhance transformer blocks with sparsely activated experts, increasing model width without a surge in computational load (Feng et al., 2024; Wu et al., 2024b; Gao et al., 2024a). These architectures vary in sampling strategies and routing mechanisms. For instance, MoRAL (Yang et al., 2024) adapts LLMs for new domains and enable them to be efficient lifelong learners. LLaVA-MoLE (Chen et al., 2024) routes tokens to domain-specific experts, improving performance over standard LoRA. LoRAMoE (Dou et al., 2024) integrates LoRAs with a routing network to prevent knowledge loss. PESC (Wu et al., 2024a) shifts dense models to sparser structures, cutting computational costs and GPU usage. MoE-LoRA (Luo et al., 2024) introduces a novel parameter-efficient MoE method with Layer-wise Expert Allocation (MoLA) for transformer-based models. MoCLE (Gou et al., 2023) proposes a MoE architecture to activate task-customized model parameters based on instruction clusters. MIXLORA (Li et al., 2024) integrates LoRAs as stochastic experts to enhance model capacity and generalization. Compared with these existing approaches, *ULoRA-MoE* is the first LoRA MoE module to enable probabilistic reconstruction, effectively capturing model uncertainty for time series anomaly detection.

## 5 EXPERIMENTS

### 5.1 EXPERIMENTAL SETTINGS.

We employ *ULoRA-MoE* on two representative time series foundation model: 1. GPT4TS (Zhou et al., 2023), a pretrained LLM model based on GPT-2, 2. UniTS, a time series foundation model pretrained by pure time series data. Both of them support anomaly detection task.

**Datasets.** To demonstrate the effectiveness of *ULoRA-MoE*, we evaluate it on five widely used multivariate real-world time series anomaly detection benchmarks: SMD (Su et al., 2019), MSL (Hundman et al., 2018), SMAP (Hundman et al., 2018), PSM (Abdulaal et al., 2021), and SWaT

| Data | Domain | Dimension | Training | Validation | Test | AR(%) |
|------|--------|-----------|----------|------------|------|-------|
| MSL | Spacecraft | 55 | 46653 | 11664 | 73729 | 10.5 |
| PSM | Server | 25 | 105984 | 26497 | 87841 | 27.8 |
| SMAP | Spacecraft | 25 | 108146 | 27037 | 427617 | 12.8 |
| SMD | Server | 38 | 566724 | 141681 | 708420 | 4.2 |
| SWaT | Water treatment | 51 | 396000 | 99000 | 449919 | 12.1 |
| Creditcard | Finance | 29 | 113923 | 28480 | 142404 | 0.17 |
| CICIDS | Web | 72 | 68092 | 17023 | 85116 | 1.28 |
| SWAN | Space weather | 38 | 48000 | 12000 | 60000 | 23.8 |

Table 1: Details of benchmark datasets for evaluation. AR (anomaly ratio) represents the abnormal proportion of the whole dataset.

| Data | MSL | | | PSM | | | SMAP | | | SMD | | |
|---|---|---|---|---|---|---|---|---|---|---|---|---|
| Metric | P | R | F1 | P | R | F1 | P | R | F1 | P | R | F1 |
| GPT4TS-FT | 60.23 | 80.23 | 68.8 | 75.44 | 69.02 | 72.09 | 55.57 | 50.23 | 52.76 | 80.61 | 80.2 | 80.4 |
| GPT4TS-LoRA | 59.65 | 66.85 | 63.04 | 75.34 | 73.87 | 74.6 | 56.82 | 50.41 | 53.42 | 80.89 | 78.29 | 79.57 |
| GPT4TS (*ULoRA-MoE*) | 63.64 | 88.73 | **74.12** | 65.67 | 87.11 | **74.89** | 61.27 | 75.11 | **67.49** | 79.66 | 85.07 | **82.28** |

Table 2: Fine-tuning performance on pretrained LLM model GPT4TS across real-world datasets (1-4). All results are presented in percentages. The best results are highlighted in bold.

| Data | Creditcard | | | CICIDS | | | SWAN | | | SWaT | | |
|---|---|---|---|---|---|---|---|---|---|---|---|---|
| Metric | P | R | F1 | P | R | F1 | P | R | F1 | P | R | F1 |
| GPT4TS-FT | 65.59 | 71.1 | 68.24 | 56.15 | 92.57 | 69.9 | 63.26 | 64.59 | 63.92 | 65.05 | 60.09 | 62.48 |
| GPT4TS-LoRA | 64.67 | 69.24 | 66.87 | 52.16 | 91.22 | 66.37 | 58.36 | 71.96 | 64.45 | 65.43 | 59.06 | 62.08 |
| GPT4TS (*ULoRA-MoE*) | 60.53 | 92.82 | **73.28** | 57.88 | 98.02 | **72.78** | 66.94 | 72.72 | **69.71** | 67.17 | 96.37 | **79.16** |

Table 3: Fine-tuning performance on pretrained LLM model GPT4TS across real-world datasets (5-8). All results are presented in percentages. The best results are highlighted in bold.

(Mathur & Tippenhauer, 2016). We also evaluate CICIDS, Creditcard, and SWAN from the NeurIPS-TS competition (Lai et al., 2021). The details of benchmark datasets are shown on Table 1.

**Baselines.** We compare *ULoRA-MoE* with existing fine-tuning approaches on time series foundation model, including GPT4TS fine tuning (GPT4TS-FT) (Zhou et al., 2023), UniTS fine tuning (UniTS-FT) and UniTS prompt tuning (UniTS-PMT) (Gao et al., 2024b). For both of the model, we also consider fine-tuning them with LoRA. Furthermore, we compare *ULoRA-MoE* with 22 baselines for comprehensive evaluations, including the linear transformation-based models: PCA (Shyu et al., 2003); the density estimation-based methods: HBOS (Goldstein & Dengel, 2012), LOF (Breunig et al., 2000); the outlier-based methods: IForest (Liu et al., 2008), LODA (Pevný, 2016); the neural network-based models: Anomaly Transformer (A.T.) (Xu et al., 2021), Autoformer (Wu et al., 2021), Crossformer (Zhang & Yan, 2023), DLinear (Zeng et al., 2023), ETSformer (Woo et al., 2022), FEDformer (Zhou et al., 2022b), FiLM (Zhou et al., 2022a), Informer (Zhou et al., 2021), iTransformer (Liu et al., 2023a), LightTS (Zhang et al., 2022), MICN (Wang et al., 2023), Pyraformer (Liu et al., 2021), Reformer (Kitaev et al., 2020), TimesNet (Wu et al., 2022), DCdetector (Yang et al., 2023), D3R (Wang et al., 2024), ModernTCN (Luo & Wang, 2024).

**Evaluation Metrics.**

For prediction accuracy, many existing methods use point adjustments (PA) to refine the detection results. However, PA assumes that if even one point in an anomaly segment is correctly detected, the entire segment is considered correctly detected, which is unreasonable (Wang et al., 2024). Recent works show that PA can lead to misleading performance evaluations (Huet et al., 2022). To address this issue, we use the affiliation-based F1 score (F1) (Huet et al., 2022), which has been widely adopted recently (Wang et al., 2024; Yang et al., 2023). This score considers the average directed distance between predicted anomalies and ground truth anomaly events to calculate affiliated precision (P) and recall (R). After obtaining the anomaly score, we use the widely adopted SPOT (Siffer et al., 2017) method, as in existing works (Wang et al., 2024; Su et al., 2019), to determine the threshold to identify outliers.

## 5.2 MAIN RESULTS.

We first compare our method, *ULoRA-MoE*, with other fine-tuning approaches on GPT4TS and UniTS. The results are shown in Table 2, Table 3, Table 4, and Table 5. It can be found that *ULoRA-MoE* significantly enhances the fine-tuning performance compared to the original deterministic fine-tuning methods. Additionally, we contrast *ULoRA-MoE* with the LoRA method to highlight that the inclusion of the MoE structure can help effectively fine-tune the model to capture the boundaries between various normal and abnormal patterns.

Moreover, we assess the fine-tuned foundation models using *ULoRA-MoE* against a comprehensive set of anomaly detection benchmarks. The findings, presented in Table 6 and Table 7, indicate that both GPT4TS (*ULoRA-MoE*) and UniTS (*ULoRA-MoE*) consistently rank as the best or second-best across 8 real-world benchmark datasets. This results shows *ULoRA-MoE* greatly improves the

| Data | MSL | | | PSM | | | SMAP | | | SMD | | |
|---|---|---|---|---|---|---|---|---|---|---|---|---|
| Metric | P | R | F1 | P | R | F1 | P | R | F1 | P | R | F1 |
| UniTS-FT | 60.13 | 78.07 | 67.94 | 62.88 | 98.95 | 76.89 | 56.65 | 51.53 | 53.97 | 76.97 | 82.08 | 79.44 |
| UniTS-PMT | 60.52 | 83.64 | 70.23 | 72.73 | 73.16 | 72.94 | 57.26 | 53 | 55.05 | 66.49 | 98.3 | 79.32 |
| UniTS-LoRA | 59.79 | 83.61 | 69.72 | 71.37 | 80.54 | 75.68 | 57.07 | 52.86 | 54.89 | 79.93 | 79.82 | 79.87 |
| UniTS (*ULoRA-MoE*) | 61.92 | 94.37 | **74.78** | 66.54 | 95.31 | **78.37** | 59.98 | 60.32 | **60.15** | 73.07 | 88.92 | **80.22** |

Table 4: Fine-tuning performance on pretrained time series foundation model UniTS across real-world datasets (1-4). All results are presented in percentages. The best results are highlighted in bold.

| Data | Creditcard | | | CICIDS | | | SWAN | | | SWaT | | |
|---|---|---|---|---|---|---|---|---|---|---|---|---|
| Metric | P | R | F1 | P | R | F1 | P | R | F1 | P | R | F1 |
| UniTS-FT | 61.98 | 66.77 | 64.29 | 53.22 | 72.65 | 61.43 | 62.78 | 56.84 | 59.66 | 67.61 | 70.24 | 68.9 |
| UniTS-PMT | 66.21 | 66.79 | 66.5 | 52.79 | 66.57 | 58.88 | 67.02 | 60.61 | 63.65 | 66.59 | 55.89 | 60.78 |
| UniTS-LoRA | 66.4 | 71.55 | 68.88 | 52.6 | 67.11 | 58.98 | 68.21 | 59.83 | 63.75 | 68.28 | 69.68 | 68.98 |
| UniTS (*ULoRA-MoE*) | 55.18 | 95.43 | **69.93** | 59.26 | 93.53 | **72.55** | 57.13 | 88.86 | **69.55** | 66.05 | 90.03 | **76.2** |

Table 5: Fine-tuning performance on pretrained time series foundation model UniTS across real-world datasets (5-8). All results are presented in percentages. The best results are highlighted in bold.

| Data | MSL | | | PSM | | | SMAP | | | SMD | | |
|---|---|---|---|---|---|---|---|---|---|---|---|---|
| Metric | P | R | F1 | P | R | F1 | P | R | F1 | P | R | F1 |
| HBOS | 53.23 | 94.44 | 68.09 | 55.06 | 83.47 | 66.35 | 39.56 | 70.31 | 50.63 | 64.03 | 39.4 | 48.79 |
| Iforest | 50.23 | 98.1 | 66.44 | 60.62 | 88.9 | 72.09 | 41.13 | 77.68 | 53.78 | 64.71 | 49.59 | 56.15 |
| LOF | 54.77 | 93.75 | 69.14 | 68.62 | 61.81 | 65.04 | 46.07 | 78.13 | 57.96 | 66.8 | 48.44 | 56.16 |
| PCA | 50.65 | 98.05 | 66.79 | 64.55 | 82.1 | 72.27 | 39.79 | 71.66 | 51.16 | 73.57 | 82.56 | 77.81 |
| LODA | 48.67 | 98.01 | 65.04 | 60.07 | 80.83 | 68.92 | 39.47 | 69.98 | 50.47 | 72.93 | 78.89 | 75.79 |
| A.T. | 50.98 | 97.09 | 66.86 | 54.35 | 85.92 | 66.58 | 43.82 | 63.93 | 52 | 69.47 | 82.06 | 75.24 |
| Autoformer | 60.28 | 83.76 | 70.11 | 90.21 | 37.9 | 53.36 | 57.03 | 52.42 | 54.63 | 88.59 | 46.74 | 61.2 |
| Crossformer | 59.57 | 83.33 | 69.48 | 77.36 | 60.08 | 67.63 | 57.16 | 52.1 | 54.51 | 86.37 | 55.62 | 67.66 |
| DLinear | 60.2 | 83.5 | 69.96 | 71.51 | 76.22 | 73.79 | 57.24 | 52.24 | 54.62 | 86.35 | 56.59 | 68.37 |
| ETSformer | 61.42 | 82.41 | 70.38 | 76.62 | 66.35 | 71.11 | 56.91 | 51.29 | 53.95 | 86.87 | 51.53 | 64.69 |
| FEDformer | 60.17 | 83.78 | 70.04 | 88.47 | 39.99 | 55.08 | 56.64 | 52.08 | 54.26 | 88.51 | 45.44 | 60.05 |
| FiLM | 60.46 | 83.58 | 70.17 | 67.36 | 72.94 | 70.04 | 56.11 | 52.36 | 54.17 | 82.73 | 56.99 | 67.49 |
| Informer | 59.48 | 83.34 | 69.42 | 78.25 | 58.74 | 67.11 | 56.4 | 51.86 | 54.03 | 88.07 | 49.12 | 63.07 |
| iTransformer | 60.7 | 84.37 | 70.61 | 69.9 | 78.73 | 74.05 | 57.9 | 53.57 | 55.65 | 85.12 | 57.92 | 68.93 |
| LightTS | 61.28 | 82.45 | 70.31 | 70.1 | 65.58 | 67.76 | 56.71 | 52.48 | 54.51 | 85.75 | 54.15 | 66.38 |
| MICN | 59.94 | 83.52 | 69.79 | 79.56 | 63.49 | 70.63 | 57.66 | 52.69 | 55.06 | 88.47 | 44.26 | 59.01 |
| Pyraformer | 59.53 | 83.39 | 69.47 | 75.21 | 65.69 | 70.12 | 57.23 | 53.16 | 55.12 | 86.24 | 52.17 | 65.01 |
| Reformer | 59.6 | 83.41 | 69.52 | 77.56 | 59.59 | 67.4 | 57.27 | 53.24 | 55.18 | 87.73 | 49.85 | 63.58 |
| TimesNet | 59.93 | 82.81 | 69.54 | 74.01 | 68.23 | 71 | 58.74 | 53.69 | 56.1 | 85.22 | 52.88 | 65.26 |
| Dcdetector | 52.17 | 97.01 | 67.85 | 52.66 | 71.1 | 63.23 | 39.82 | 72.73 | 51.47 | 51 | 95.22 | 66.42 |
| D3R | 52.25 | 67.07 | 58.74 | 74 | 72.7 | 73.35 | 56.04 | 45.6 | 50.28 | 60.43 | 71.2 | 65.37 |
| ModernTCN | 60.6 | 84.24 | 70.49 | 71.37 | 73.69 | 72.51 | 58.18 | 53.15 | 55.55 | 78.74 | 83.36 | 80.98 |
| GPT4TS (*ULoRA-MoE*) | 63.64 | 88.73 | 74.12 | 65.67 | 87.11 | 74.89 | 61.27 | 75.11 | **67.49** | 79.66 | 85.07 | **82.28** |
| UniTS (*ULoRA-MoE*) | 61.92 | 94.37 | **74.78** | 66.54 | 95.31 | **78.37** | 59.98 | 60.32 | 60.15 | 73.07 | 88.92 | 80.22 |

Table 6: Comparison of the fine-tuned time series foundation model using *ULoRA-MoE* against existing baselines across real-world datasets (1-4). All results are presented in percentages. The higher values for all metrics represent the better performance. The best affiliated F1 scores are highlighted in bold. The second-best affiliated F1 scores are underlined.

fine-tuning performance on time series foundation models and achieves state-of-the-art performance on anomaly detection.

## 5.3 ABLATION STUDY.

We conducted an ablation study to compare *ULoRA-MoE* with the deterministic MoE-LoRA. The only difference is that MoE-LoRA follows the existing approaches to use mean squared error as the anomaly score. The comparison results are depicted in Figure 2. It shows that *ULoRA-MoE* consistently achieves significant improvements over the deterministic MoE-LoRA across 8 real-world datasets, with an average improvement of 15% on GPT4TS and 13.8% on UniTS. These results validate the effectiveness of employing an uncertainty-aware anomaly score function to mitigate the detrimental effects of anomaly contamination.

| Data | Creditcard | | | CICIDS | | | SWAN | | | SWaT | | |
|---|---|---|---|---|---|---|---|---|---|---|---|---|
| **Metric** | P | R | F1 | P | R | F1 | P | R | F1 | P | R | F1 |
| HBOS | 64.86 | 40.64 | 49.97 | 54.03 | 47.73 | 50.69 | 84.37 | 24.99 | 38.56 | 60.31 | 75.07 | 66.88 |
| Iforest | 61.57 | 37.47 | 46.59 | 53.77 | 49.5 | 51.55 | 78.12 | 26.41 | 39.48 | 61.3 | 74.54 | 67.27 |
| LOF | 51.76 | 29.13 | 37.28 | 51.18 | 56.93 | 53.9 | 74.67 | 29.24 | 42.02 | 70.83 | 49.42 | 58.22 |
| PCA | 58.05 | 36.83 | 45.07 | 53.95 | 57.43 | 55.64 | 82.52 | 24.54 | 37.83 | 62.31 | 64.94 | 63.6 |
| LODA | 67.22 | 41.04 | 50.97 | 49.33 | 54.66 | 51.86 | 41.97 | 26.22 | 32.28 | 58.81 | 76.2 | 66.38 |
| A.T. | 52.6 | 77.46 | 62.65 | 55.72 | 28.2 | 37.45 | 58.78 | 8.68 | 15.13 | 54.12 | 98.34 | 69.82 |
| Autoformer | 67.58 | 71.92 | 69.68 | 54.42 | 56.55 | 55.47 | 77.96 | 25.85 | 38.82 | 68.06 | 71.04 | 69.52 |
| Crossformer | 66.38 | 71.02 | 68.62 | 52.24 | 69.78 | 59.75 | 84.63 | 22.42 | 35.44 | 67.73 | 68.08 | 67.9 |
| DLinear | 66.95 | 71.88 | 69.33 | 53.62 | 67.97 | 59.95 | 78.88 | 25.87 | 38.96 | 64.81 | 56.8 | 60.54 |
| ETSformer | 67.68 | 72.19 | 69.86 | 53.36 | 54.65 | 54 | 73.8 | 27.27 | 39.82 | 68.29 | 62.63 | 65.34 |
| FEDformer | 67.63 | 73.2 | 70.31 | 54.21 | 55.4 | 54.8 | 83.65 | 21.91 | 34.73 | 68 | 70.81 | 69.38 |
| FiLM | 57.72 | 61.1 | 59.36 | 52.45 | 39.93 | 45.34 | 87.87 | 19.8 | 32.32 | 62.41 | 58.62 | 60.45 |
| Informer | 66.81 | 71.34 | 69 | 53.7 | 57.03 | 55.32 | 75.76 | 30.14 | 43.12 | 68.64 | 73.11 | 70.8 |
| iTransformer | 65.87 | 70.97 | 68.32 | 55.07 | 52.64 | 53.82 | 77.71 | 22.25 | 34.6 | 64.7 | 59.75 | 62.13 |
| LightTS | 65.56 | 72.87 | 69.02 | 54.12 | 48.72 | 51.28 | 82.41 | 20.55 | 32.89 | 65.99 | 71.5 | 68.63 |
| MICN | 67.69 | 72.91 | 70.2 | 52.77 | 56.06 | 54.37 | 66.2 | 32.19 | 43.32 | 67.58 | 69.86 | 68.7 |
| Pyraformer | 67.07 | 71.66 | 69.29 | 53.79 | 55.91 | 54.83 | 81.51 | 25.18 | 38.47 | 65.88 | 75.86 | 70.52 |
| Reformer | 67.78 | 73.33 | 70.45 | 52.46 | 60.33 | 56.12 | 74.72 | 32.21 | 45.01 | 68.13 | 73.81 | 70.86 |
| TimesNet | 65.15 | 70.01 | 67.5 | 56.21 | 50.14 | 53 | 82.55 | 20.53 | 32.88 | 65.79 | 58.52 | 61.94 |
| Dcdetector | 48.73 | 70.12 | 57.5 | 51.06 | 29.67 | 37.53 | 43.2 | 6.49 | 11.28 | 52.15 | 98.38 | 68.17 |
| D3R | 63.55 | 74.26 | 68.49 | 45.98 | 99.94 | 62.99 | 63.87 | 40.31 | 49.43 | 62.89 | 78.86 | 69.97 |
| ModernTCN | 66.64 | 72.05 | 69.24 | 54.75 | 49.07 | 51.75 | 72.07 | 33.36 | 45.61 | 66.31 | 61.45 | 63.79 |
| GPT4TS (*ULoRA-MoE*) | 60.53 | 92.82 | **73.28** | 57.88 | 98.02 | **72.78** | 66.94 | 72.72 | **69.71** | 67.17 | 96.37 | **79.16** |
| UniTS (*ULoRA-MoE*) | 55.18 | 95.43 | 69.93 | 59.26 | 93.53 | 72.55 | 57.13 | 88.86 | 69.55 | 66.05 | 90.03 | 76.2 |

Table 7: Comparison of the fine-tuned time series foundation model using *ULoRA-MoE* against existing baselines across real-world datasets (5-8). All results are presented in percentages. The higher values for all metrics represent the better performance. The best affiliated F1 scores are highlighted in bold. The second-best affiliated F1 scores are underlined.

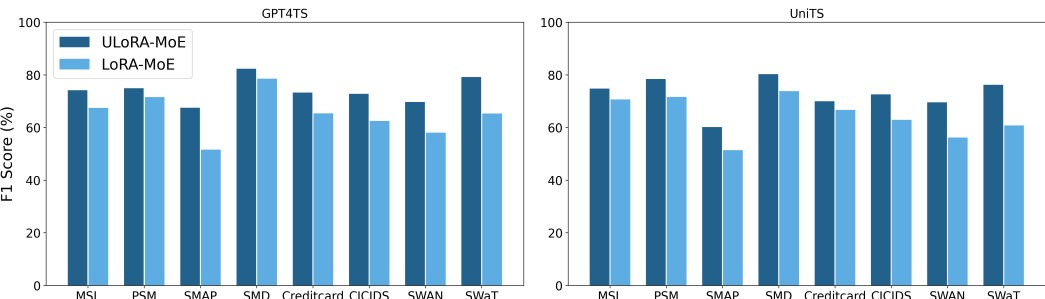

Figure 2: Comparison of *ULoRA-MoE* with deterministic LoRA-MoE for time-series anomaly detection across 8 real-world datasets. The higher values represent the better performance.

## 6 CONCLUSION, LIMITATION, AND FUTURE WORK

In this paper, we propose *ULoRA-MoE*, a probabilistic fine-tuning framework for time series foundation models targeting anomaly detection. *ULoRA-MoE* leverages a resource-efficient Mixture-of-Experts (MoE) model with LoRA to precisely delineate the boundaries between normal and anomalous data. Additionally, we utilize the Gumbel-Softmax distribution for categorical sampling on the MoE router to estimate the epistemic uncertainty of the fine-tuned foundation model. Given the estimated uncertainty, we propose a score function for anomaly detection to calibrate the fine-tuned model in the presence of contaminated training data. Our evaluations demonstrate that *ULoRA-MoE* is effective with both LLM-based and time-series-based foundation models, achieving state-of-the-art performance on a broad spectrum of time series anomaly detection benchmarks. While *ULoRA-MoE* is applicable to all time series foundation models using transformer architectures, practical challenges due to their well-encapsulated nature have limited our current implementation. Addressing these challenges remains an area for future research. Furthermore, while this work focuses primarily on anomaly detection, we aim to extend *ULoRA-MoE* to other time series modeling tasks, including forecasting, imputation, and classification. For instance, *ULoRA-MoE* is a promising approach for probabilistic forecasting.

REPRODUCIBILITY STATEMENT

The experimental details are shown in Appendix A.1. We will release the code upon legal approval.

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

# A   APPENDIX

## A.1   IMPLEMENTATION DETAILS

In our experiments, we use a sliding window of $96$ and a step size of $48$ across all datasets. We employ grid search to determine the optimal SPOT parameters for each dataset and record the results that yield the highest affiliated F1 scores. *ULoRA-MoE* is trained with a total of $5$ LoRA experts, generating $5$ samples to compute the anomaly score from the Gumbel-Softmax distribution. Each LoRA expert has a rank of $8$. We train our model on NVIDIA A100 GPU.

