# OpenReview forum: "Uncertainty-aware Fine-tuning on Time Series Foundation Model for Anomaly Detection"
_ICLR.cc/2025/Conference — ICLR 2025 Conference Withdrawn Submission_

### Official Review · Reviewer_CE5n · 2024-10-26

**Soundness:** 1
**Presentation:** 2
**Contribution:** 1
**Rating:** 1
**Confidence:** 4

**Summary:**

The authors propose an uncertainty-aware fine-tuning approach ULoRA-MoE for anomaly detection using the foundation model, MoE and LoRA, and think that ULoRA-MoE delivers better performance by capturing the boundaries of various types of normal and anomalous patterns.

**Strengths:**

1. The author considers the impact of the anomalies in the dataset for fine-tuning the foundation model for anomaly detection with MoE.
2. The authors propose to fine-tune the foundation model for anomaly detection with LoRA.
3. The authors show experimental results on 8 datasets.

**Weaknesses:**

1. The paper directly uses several existing techniques, such as MoE, LoRA, probability modeling and Gumbel-Softmax, into existing time series foundation models for fine-tuning. The contributions and novelty are limited.

2. There are no experiments to show or theoretical proof of why MoE and LoRA can learn the boundaries of normal and anomalous patterns. More experiments or analyses would help demonstrate how or why MoE and LoRA learn these boundaries.

3. More experiments or analyses would help demonstrate why probability modeling can solve the anomalies in the dataset. However, I do not think it can solve the challenges in this paper. The probability modeling anomaly scores in this paper are also optimized for the anomalies in the dataset. I believe it cannot solve the challenges in this paper.

4. The calculations of MoE and a set of reconstruction samples are much more complex than the baselines in both time and space complexity. No analysis is conducted on efficiency and effectiveness. More experiments or analyses would help demonstrate the tradeoff.

5. As the paper uses the probability of a set of reconstruction samples as the anomaly score as contributions, the authors should also include related works of existing probability models and compare them by novelty and effectiveness, such as [1] and [2]. The authors should also include a specific comparison section that analyzes how their approach differs from and potentially improves upon these existing probabilistic methods for time series analysis.

    [1] CSDI: Conditional Score-based Diffusion Models for Probabilistic Time Series Imputation. Yusuke Tashiro et al., NeurIPS 2021.

    [2] Diffusion-TS: Interpretable Diffusion for General Time Series Generation, Xinyu Yuan et al., ICLR 2024.

6. More benchmarks, such as UCR [3], and metrics, such as PR-AUC and ROC-AUC, should be considered. The authors should justify their choice of benchmarks and metrics, and explain why they believe their current selections are sufficient or how they plan to expand their evaluation.

    [3] Current Time Series Anomaly Detection Benchmarks are Flawed and are Creating the Illusion of Progress. Eamonn J. Keogh et al., TKDE 2023.

7. No codes available. I find the results in this paper are not reliable. From the original paper such as their baseline D3R, the F1 results are 0.7609, 0.8682, and 0.7812 for PSM, SMD, and SWaT, which are much larger than the results in this paper. Can you explain what is the reason and provide the codes to evaluate your results and ensure reproducibility?

8. The titles are different on the PDF and submission page. The authors should ensure consistency between the title on the PDF and the submission page, and clarify which title they intend to use.

9. Many writing errors need to be improved, such as line 88 $X$ should be $\hat{X}$.

**Questions:**

see weaknesses.

---

### Official Review · Reviewer_2i4b · 2024-10-31

**Soundness:** 1
**Presentation:** 2
**Contribution:** 1
**Rating:** 3
**Confidence:** 4

**Summary:**

Existing foundation time series models struggle to deliver great performance in anomaly detection tasks, primarily because their unsupervised paradigm is prone to contamination by potential anomalies within the training data. Additionally, current methods fall short in capturing the boundaries of various types of normal and anomalous patterns. To address these challenges, the paper proposes a general uncertainty-aware fine-tuning approach using resource-efficient MoE model based on LoRA.

**Strengths:**

- The paper considers the impact of potential anomalies in the dataset, as well as the shortcomings of existing models in capturing the boundaries of various types of normal and anomalous patterns.
- The paper proposes a fine-tuning approach based on LoRA, which is helpful for two general types of time series foundation models in anomaly detection tasks.

**Weaknesses:**

- The paper is poorly written, and often lacks intuitions and justifications. The paper needs clear motivations on why use MoE and LoRA, and needs clarifications on how MoE and LoRA solve their challenges. There are also issues with symbol definitions, such as the repeated use of \hat{x}, which creates reading obstacles.

- The paper simply combines several existing fine-tuning techniques into existing foundation models without novel contributions. The MoE and LoRA seem not to have relevance to the boundaries of normal and anomalous patterns. The anomaly score function is applied during the inference stage and thus it cannot address the impact of anomalous data during the fine-tuning process.

- The main claims of the paper are not adequately supported with evidence. The paper needs to provide discussion and analyses on why the proposed method can address the challenges mentioned in the introduction and how it effectively tackles them. For example, why and how the sampling-based routing strategy quantify the epistemic uncertainty?

- I find it difficult to believe that various modules proposed by the authors are effective and there are no codes for evaluation. The experiments are incomplete, and the analysis is inadequate. The authors merely list the performance of the model without validating the effectiveness of individual components, such as the Gumbel-Softmax sampling approach and the proposed anomaly score. Simply stating that the methods is SOTA is insufficient. I would prefer to understand which part plays a critical role and how they impacted the results. More insight analysis should be provided, including but not limited to thorough ablation studies, hyperparameter analysis, and visualization analysis.

- The paper should consider additional metrics to provide a more comprehensive evaluation of the model. Besides, using only two foundational models is insufficient. More models, such as Moment[1] and Timer[2], should be considered.

[1] Goswami M, Szafer K, Choudhry A, et al. Moment: A family of open time-series foundation models. ICML'24.
[2] Liu Y, Zhang H, Li C, et al. Timer: Generative Pre-trained Transformers Are Large Time Series Models. ICML'24.

**Questions:**

- Please address weaknesses

- Why and how the sampling-based routing strategy quantify the epistemic uncertainty?

- In line 62, How does the model utilizes resource-efficient MoE module to learn different types of anomalies and accurately define the boundary between each type of the normal and anomalous data?

- If the NLL anomaly score is used during the inference stage, then how does it calibrate the fine-tuning process? As described in line 137, in fine-tuning stage, the model still will be compromised by these anomaly contamination.

- How does the NLL penalizes predictions with high epistemic uncertainty?

- If the NLL anomaly score is effective, could other reconstruction-based methods also see performance improvements by using the NLL anomaly score?

---

### Official Review · Reviewer_5hqF · 2024-10-31

**Soundness:** 2
**Presentation:** 3
**Contribution:** 2
**Rating:** 5
**Confidence:** 4

**Summary:**

The authors propose ULoRA-MoE, a method to fine-tune a time series foundation model for the time series anomaly detection task. The fine-tuned method is based on a MoE LoRA (Li et al. 2024, Wu et al. 2024) (MoE = mixture of experts).

The main contribution is related to the *management of the uncertainty in the predictions*: the authors propose a novel routing mechanism through Gumbel-Softmax gates (Maddison et al. 2016) (instead of softmax gates), that allows the generation of multiple reconstruction samples, i.e. to get an empirical distribution of the predictions. The final anomalous scoring is driven by both the reconstruction error (larger anomalous score for large errors) and the uncertainty (larger anomalous score for larger uncertainty).

In the experiments, the fine-tuned methodology is applied on GPT4TS and UniTS foundation models. Two rounds of experiments are performed: 1. against the original foundational models or existing fine-tuning methods of those models (especially against LoRA-MoE), and 2. against the other anomaly detection algorithms (isolation forest, Loda, deep learning models, etc.). 8 (source of) datasets are used, and the evaluation is performed against the affiliation F1-score.

**Strengths:**

- The uncertainty mechanism: clever idea by combining MoE LoRA with the Gumbel-Softmax gates; in addition ablation in this dimension is performed against the LoRA-MoE and is convincing. I agree that this is a way to find the boundaries between normal and abnormal patterns.

- The performance improvement against the numerous tested sota models is large. The selected metric is valid (instead of using point-adjust metrics).

**Weaknesses:**

- The claim regarding the contamination. Authors explain in the abstract that the "effectiveness in anomaly detection is often inferior (...) due to anomaly contamination in the training data". The relation between performance and contamination is difficult to assess based on the experiments performed by the authors. For instance, SWaT is not contaminated (as explained by the provided reference Xu et al.), while the performance with ULoRA-MoE is still better. I suggest, if this is a claim by the authors: (a) to provide an estimated contamination for each dataset, (b) to perform an ablation with increasing value of contamination (maybe following Xu et al. on Epilespy and DSADS?)

- Availability of the code/results: is there a confirmation that the code will be provided? In addition, it would be preferable to include the pre-computed anomaly scores for each (method, dataset) of the tables, so that other researchers can directly check for other thresholds/metrics.

- Questions/comments below are related to some missing important details that impact the current rating and soundness assessments.

**Questions:**

- How the thresholds have been selected for all other methods but yours? The results are quite low for many algorithms. For instance, with random numerous predictions, we would expect an affiliation F1-score close to 0.66 (precision of 0.5 since those are random predictions, and recall close to 1 since those predictions are numerous).

- Equation 6: I don't get why x seems to be a single numerical value, instead of a vector. It should also be a mean vector and a variance-covariance matrix. In that case, how this matrix is estimated with 5 generated samples (for instance with SWaT which contains a large number of features)?

- Equation 6: this is the likelihood for the Gaussian distribution, how does this apply in your case?

- There are issues known by the community in some of the datasets, in particular PSM, SMAP, SMD, and SWaT. Those issues should be at least mentioned (see references Wu et Keogh (2021) and TimeSeAD (Wagner et al. 2023) for details).

Minor or other comments:
- Since SMD contains many different datasets, how the final F1 score has been computed?
- line 197: what is a PLoRA-MoE?
- typos: "It incorporate LoRA into MoE involves applying low-rank updates"; boundries
- It could be also interesting to quantify this assertion: "Each expert module of MoE can help learn different types of anomalies". This is intuitively coherent, but actual evidence are missing in the paper.
- Many ablations could be interesting: change the number or rank of the LoRA experts, study the distribution of the generated samples, instead of generating only 5 samples. I'm not suggesting to perform them during the revision stage.

---

### Official Review · Reviewer_shSX · 2024-11-01

**Soundness:** 2
**Presentation:** 3
**Contribution:** 3
**Rating:** 6
**Confidence:** 3

**Summary:**

This paper introduces ULoRA-MoE, a fine-tuning approach for time-series foundation models in anomaly detection. By integrating LoRA and Mixture-of-Experts (MoE), it adapts the pre-trained models to learn diverse normal and abnormal patterns efficiently. An uncertainty-aware router using Gumbel-Softmax enables selective sampling, improving model stability and accuracy. To handle the uncertainty and anomaly contamination, It also incorporate negative log-likelihood as the anomaly score. Extensive experiments show that ULoRA-MoE outperforms existing methods across eight real-world benchmarks.

**Strengths:**

1.	The paper is well-structured, providing a clear explanation of each component, including the use of techniques like MoE and LoRA for fine-tuning time-series foundation models. The motivation and discussion are straightforward and easy to follow.
	2.	It introduces an innovative routing mechanism using Gumbel-Softmax for efficient sampling, which reduces memory usage and computation time while improving training efficiency.
	3.	The study includes extensive comparisons across multiple datasets and a broad set of baseline models.

**Weaknesses:**

1.	Much of the methodology follows standard approaches for fine-tuning foundation models (e.g., LoRA and MoE). The uncertainty-aware component based on negative log-likelihood appears limited, as it mainly penalizes highly uncertain predictions. Exploring alternative distributions and more advanced uncertainty quantification methods could strengthen this aspect.
	2.	The fine-tuning is only tested on GPT4TS and UniTS models, omitting other popular foundation models like Lag-Llama, TimesFM, and TimesGPT, which limits the generalizability and impact of the approach.
	3.	While the authors claim effective epistemic uncertainty quantification, the proposed approach primarily involves sampling and a negative likelihood score. A deeper investigation into producing confidence bands or capturing anomaly score distributions would better support this claim.

**Questions:**

Did you explore alternative methods for uncertainty quantification beyond a negative log-likelihood penalty? For example, have you considered using alternative distribution functions or confidence interval approaches to deepen the analysis of uncertainty?

Could you elaborate on how the approach you proposed captures the epistemic uncertainty across different pre-trained models?

Given that Lag-Llama, TimesFM, and TimesGPT are widely used in time-series applications, could you discuss why these models were not included in the experiments?

---

### Official Review · Reviewer_cLjo · 2024-11-03

**Soundness:** 3
**Presentation:** 3
**Contribution:** 3
**Rating:** 5
**Confidence:** 5

**Summary:**

The gist of the paper is that anomaly detectors aren't suitable because training (or fine-tuning) data can be contaminated with anomalies. That's why the authors proposed to have models of experts each of which captures different types of anomalies. The authors propose to quantify the epistemic uncertainty to calibrate the fine-tuning of pre-trained transformer-based models to minimize the contamination from the data points in the fine-tuning set. The authors here use LoRA MoE to sample at each layer with LoRA module instead of sampling in all the parameter space, which gives space for computational boost.

**Strengths:**

1. Kudos to the authors by comparing with traditional baselines like IForest, LOF, HBOS. Although the literature seems to have forgotten about them, most of the time an IForest or a LOF beat current "SoTA" methods.
2. The experiments are extensive and thorough clearly showing ULoRA-MoE's integration advantage in transformer networks.
3. Props to using the "adjusted" F1 score instead of PA which overestimates the performances of anomaly detectors. I'm curious how the performance would be - maybe you can show for one dataset - if you had used PA.

**Weaknesses:**

1. Ablation study on the number of LoRA experts is missing. What happens when this increases/decreases instead of being fixed at 5.
2. The authors claim that most anomaly detectors aren't suitable because the training data might be contaminated. They also state that they are in a self-supervised paradigm. Taking into consideration these two statements, most anomaly detectors rely on the LPUE [4] technique and train on only normal data to build a notion of what normality is. Then, they test on both normal and abnormal instances and measure their performances. For fine-tuning where the fine-tuning data might be contaminated with anomalies, I get ULoRA-MoE's contribution. But for fine-tuning data where the contamination is negligible, why wouldn't a pre-trained anomaly detector be sufficient and suitable? (see Question 4).
3. In lines 58-60, the authors claim that anomaly detectors aren't able to have boundaries between anomalies and normal data (whatever "boundaries" means here...). However, although in video anomaly detection, MoCoDAD [3] doesn't need to have knowledge about anomalies whatsoever - again by relying on LPUE. The authors of this paper tackle this "lack of knowledge for anomalies" by conditioning their proposed diffusion-based model with past observations. The gist of the paper is that if the past was indeed abnormal, then the future is most probably anomalous as well and they show it by observing higher reconstruction errors.
4. The authors claim that ULoRA-MoE learns different boundaries of anomalies vs. normalities and that each MoE can capture different types of anomalies. However, the authors fail to show what kind of boundaries are we talking about here. Can these boundaries be used to actually subcategorized what anomaly is happening at timestamp $t$? How come these boundaries are better than just having a normalcy manifold and everything falling outside of this manifold is considered an anomaly? What are these boundaries? How are they traced? Via the proposed epistemic uncertainty?

Minor concerns:

5. Why don't you compare with HypAD [1] and BAE [2], although a bit old, which are relevant to your research?


Missing references:

*Uncertainty estimation for free in the hyperbolic space* $\rightarrow$ [1] Flaborea A, Prenkaj B, Munjal B, Sterpa MA, Aragona D, Podo L, Galasso F. Are we certain it's anomalous?. In Proceedings of the IEEE/CVF Conference on Computer Vision and Pattern Recognition 2023 (pp. 2897-2907).

*MoE related (boosting-based mechanism)* $\rightarrow$ [2] Sarvari H, Domeniconi C, Prenkaj B, Stilo G. Unsupervised boosting-based autoencoder ensembles for outlier detection. In Pacific-Asia Conference on Knowledge Discovery and Data Mining 2021 May 9 (pp. 91-103). Cham: Springer International Publishing.

[3] Flaborea A, Collorone L, Di Melendugno GM, D'Arrigo S, Prenkaj B, Galasso F. Multimodal motion conditioned diffusion model for skeleton-based video anomaly detection. InProceedings of the IEEE/CVF International Conference on Computer Vision 2023 (pp. 10318-10329).

[4] Zhang B, Zuo W. Learning from positive and unlabeled examples: A survey. In2008 International Symposiums on Information Processing 2008 May 23 (pp. 650-654). IEEE.

**Questions:**

1. Can you show the "adjusted" AUCPR of the models on all datasets?
2. Can you show the error bars in Figure 2?
3. How did you choose the rank of the LoRA expert (Sec. A.1)?
4. How does the contamination percentage in the fine-tuning data affect ULoRA-MoE's usefulness w.r.t. pre-trained UniTS without ULoRA-MoE for example? You can show this via a synthetic dataset that has various contamination percentages, or you can take one of the real-world datasets - say SWaT - and randomly remove normal data to obtain different anomaly percentages and show how the performances differ. I would expect that the lower the anomaly contamination, the lower the effect of ULoRA-MoE (i.e., similar adjusted F1 scores for UniTS w/ ULoRA-MoE and UniTS w/o ULoRA-MoE). I am aware that the 8 datasets used here have different anomaly percentages, but a more principled ablation study like the one described here would make the contribution of this paper more solid and fairer to the authors' claims. Can you show this, please?

---

### Note · Authors · 2024-11-13

**Comment:**

Dear Program Committee and Reviewers,

After careful consideration, we have decided to withdraw our paper from the conference review process. While we appreciate the valuable feedback provided by the reviewers, we believe that addressing these comments thoroughly will require additional time and resources to improve the quality of our work beyond the rebuttal phase.

We would like to express our gratitude for the constructive feedback, which has provided valuable insights into how we can enhance our research. We look forward to carefully implementing these suggestions and potentially resubmitting an improved version to a future venue.

Thank you once again for your time and consideration.

**Withdrawal Confirmation:**

I have read and agree with the venue's withdrawal policy on behalf of myself and my co-authors.